# Serum APOA4 Pharmacodynamically Represents Administered Recombinant Human Hepatocyte Growth Factor (E3112)

**DOI:** 10.3390/ijms22094578

**Published:** 2021-04-27

**Authors:** Sotaro Motoi, Mai Uesugi, Takashi Obara, Katsuhiro Moriya, Yoshihisa Arita, Hideaki Ogasawara, Motohiro Soejima, Toshio Imai, Tetsu Kawano

**Affiliations:** 1Eisai Product Creation Systems, KAN Product Creation Unit, Eisai Co., Ltd., 5-1-3 Tokodai, Tsukuba, Ibaraki 3002635, Japan; sotaro.motoi@gmail.com (S.M.); m-soejima@kan.eisai.co.jp (M.S.); 2KAN Research Institute, Inc., 6-8-2 Minatojima-Minamimachi, Chuo-Ku, Kobe, Hyogo 6500047, Japan; k3-moriya@hhc.eisai.co.jp (K.M.); y-arita@kan.eisai.co.jp (Y.A.); h-ogasawara@kan.eisai.co.jp (H.O.); t-imai@kan.eisai.co.jp (T.I.); 3Medicine Creation, Neurology Business Group, Translational Medicine Department, Eisai Co., Ltd., 5-1-3 Tokodai, Tsukuba, Ibaraki 3002635, Japan; m-uesugi@hhc.eisai.co.jp (M.U.); t-obara@hhc.eisai.co.jp (T.O.)

**Keywords:** rh-HGF, APOA4, ALF, c-Met, Fas

## Abstract

Background: Hepatocyte growth factor (HGF) is an endogenously induced bioactive molecule that has strong anti-apoptotic and tissue repair activities. In this research, we identified APOA4 as a novel pharmacodynamic (PD) marker of the recombinant human HGF (rh-HGF), E3112. Methods: rh-HGF was administered to mice, and their livers were investigated for the PD marker. Candidates were identified from soluble proteins and validated by using human hepatocytes in vitro and an animal disease model in vivo, in which its c-Met dependency was also ensured. Results: Among the genes induced or highly enhanced after rh-HGF exposure in vivo, a soluble apolipoprotein, *Apoa4*, was found to be induced by rh-HGF in the murine liver. By using primary cultured human hepatocytes, the significant induction of human APOA4 was observed at the mRNA and protein levels, and it was inhibited in the presence of a c-Met inhibitor. Although mice constitutively expressed *Apoa4* mRNA in the small intestine and the liver, the liver was the primary organ affected by administered rh-HGF to strongly induce APOA4 in a dose- and c-Met-dependent manner. Serum APOA4 levels were increased after rh-HGF administration, not only in normal mice but also in anti-Fas-induced murine acute liver failure (ALF), which confirmed the pharmacodynamic nature of APOA4. Conclusions: APOA4 was identified as a soluble PD marker of rh-HGF with c-Met dependency. It should be worthwhile to clinically validate its utility through clinical trials with healthy subjects and ALF patients.

## 1. Introduction

Hepatocyte growth factor (HGF) was originally discovered in the serum of patients with fulminant hepatitis due to its strong capacity to induce hepatocyte proliferation [1,2,3]. Since its discovery, vigorous research has been conducted on HGF world-wide, which revealed its pleiotropic activities [4,5,6], among which its strong and rapid anti-apoptotic effects against murine liver injury led to the dynamic investigation into a fatal and disastrous condition, acute liver failure. Although, to date, liver transplantation is the only curative treatment for the disease, the constant need for a novel drug that can help avoid or at least delay the need of organ transplantation is still unmet [7,8,9]. With the support of the Japan Agency for Medical Research and Development (AMED), we successfully manufactured a recombinant human HGF (rh-HGF) that is currently under evaluation in a phase I study with healthy Japanese volunteers (ClinicalTrials.gov Identifier: NCT03922633). During the process of drug development, it is valuable to evaluate the pharmacodynamic activity of new investigational drugs in healthy volunteers prior to evaluating its clinical efficacy or benefits for diseased conditions. Therefore, we aimed to identify a pharmacodynamic marker for rh-HGF that directly and biologically reflects exogenously administered rh-HGF, in terms of HGF-mediated c-Met receptor activation. DNA microarray analyses of the livers of rh-HGF-administered normal mice showed an induction of various genes as candidate PD markers. Among those, APOA4, which has been previously known as one of the apolipoproteins, was found to be a soluble PD marker and can be secreted into the blood as a result of HGF-c-Met signaling. Recently, it was reported that APOA4 is not just a constituent of high-density lipoprotein, but it also exerts anti-oxidant and anti-inflammatory effects to protect from liver damage in mice [10,11]. Therefore, validation of the utility of APOA4 as a PD marker of rh-HGF in clinical trials would help pharmacodynamic evaluation of rh-HGF, and furthermore, may give clues to elucidate its mechanism of action in humans, not just by direct effect on hepatocytes but also via APOA4 protein.

## 2. Results

Primarily, we tried to identify candidate PD markers for rh-HGF in vivo because it may represent the bio-distribution of the injected rh-HGF, which might mimic those in humans under the clinical setting. To obtain markers that could be easy to measure at bedside, candidates were simultaneously narrowed down to extracellular molecules or cell-surface molecules. The resulting candidates were further examined with human hepatocytes to evaluate their c-Met dependency by a c-Met inhibitor, PF-04217903, at the mRNA and protein levels. The likelihood of being a PD marker was further validated in vivo in mice. Finally, the utility of the marker was evaluated in mice with ALF, which was induced by Fas-mediated massive apoptotic events in liver cells.

### 2.1. Induction of Hepatic mRNA Expression after rh-HGF Exposure

The murine livers were harvested to collect hepatic mRNAs at 8 h or 24 h after the single rh-HGF injection (1.5 mg/kg), and the prepared cDNAs were applied to the DNA microarray according to the manufacturer’s instructions. The obtained gene list was narrowed down by filtering with molecular annotations such as “Extracellular” and/or “Plasma membrane”; thereafter, they were sorted in descending order of change in expression at 24 h (Table 1). Among them, we chose to investigate *Apoa4* due to its sustained expression up to 24 h. While the signal values of *Apoa4* at baseline and with PBS injection were around 2500, it was significantly enhanced by rh-HGF with more than a 6.46- or 5.52-fold change at 8 h and 24 h, respectively. Although there were other soluble proteins, such as *Pcsk9*, *Tnfrsf9*, and *Il6ra* (Table 1) and other growth factors such as *Ctgf* (data not shown), none of them showed a good expression magnitude and sustained expression compared to *Apoa4*.

### 2.2. Induction of APOA4 in Human Hepatocytes

APOA4 is conserved well in humans. In order to validate whether human APOA4 respond to rh-HGF, primary cultured human hepatocytes were stimulated by rh-HGF. Notably, rh-HGF induced *APOA4* mRNA (Figure 1A) and stimulated the production of APOA4 protein (Figure 1B,C) into the supernatants in a dose-dependent manner. A tyrosine receptor kinase, c-Met, is the sole receptor for HGF [6]. In order to evaluate whether APOA4 was indeed induced by c-Met mediated signaling elicited by rh-HGF [6], human hepatocytes were stimulated by rh-HGF in the presence of the c-Met inhibitor, PF-04217903 (100 nM). Inhibition of c-Met signaling significantly suppressed the induction of *APOA4* mRNA (Figure 1A) as well as production of APOA4 protein into the supernatants (Figure 1B,C). Considering the inter-species consistency in response of APOA4 to rh-HGF, APOA4 is strongly assumed to represent the pharmacodynamic nature of rh-HGF.

### 2.3. Induction of APOA4 in the Livers of Normal Mice

The liver and small intestine of mice are known to express *Apoa4*. To confirm the primary site of action of rh-HGF, the mRNA inductions of *Apoa4* in both organs were evaluated using real-time PCR. Although mRNA was detected in both the organs at baseline, the dose-dependent induction of *Apoa4* mRNA at 8 h was observed only in the liver and not in the small intestine (Figure 2). In addition, the expression in the liver was sustained at 24 h and further increased at 5 mg/kg dose (Figure 2).

### 2.4. c-Met Mediated Induction of Murine APOA4 by rh-HGF In Vivo

To confirm whether the induction of murine APOA4 by rh-HGF was certainly mediated by c-Met signaling in in vivo settings, mice were pretreated with an oral administration of a c-Met inhibitor, PF-04217903 (50 mg/kg), thereafter, rh-HGF was administered. Consistent with the results of human hepatocytes, the dose-dependent elevation of *Apoa4* mRNA by rh-HGF was strongly suppressed by c-Met inhibition (Figure 3A). The dose-dependent increase in serum APOA4 protein was also significantly diminished by c-Met signal inhibition (Figure 3B,C).

### 2.5. Pharmacodynamic Response of APOA4 in Murine ALF

To validate the utility of APOA4 in liver injury, murine APOA4 was measured in the serum obtained from murine ALF that was developed by Fas-mediated liver cell apoptosis [5,12]. We previously reported that rh-HGF significantly suppresses the elevation of aspartate aminotransferase (AST) and alanine transaminase (ALT) released from the damaged liver cells and prolongation of the prothrombin time (PT) due to the preservation of a functional liver mass [13]. By using exactly the same experimental protocol as we previously reported [13], we measured serum APOA4 in murine ALF. Under ALF condition, serum APOA4 was elevated over that of normal mice (Figure 4A; rh-HGF, 0 mg/kg) and it was further and significantly increased by rh-HGF administration (Figure 4).

## 3. Discussion

There has been substantial progress in medical practice for the care of ALF patients by incorporating an etiology-based approach and artificial liver support. Among these, liver transplantation is the most reliable measure to rescue potentially fatal cases, especially comatose patients with ALF [8,14,15]. However, it is also true that transplantation is not always applicable for the patients who have an indication of liver transplantation, primarily due to the shortage of donors. Therefore, a new treatment option is needed to prevent progressive liver damage, preserve residual mass of functional liver, and promote the repair and regeneration of hepatocytes. HGF, due to its strong anti-apoptotic effect, stimulation of tissue repair, and regeneration activities, is the most promising hope to address this long-standing problem [4,5,16]. We successfully manufactured rh-HGF that pre-clinically showed substantial inhibition of liver damage with the strong preservation of prothrombin time in a Fas-triggering murine ALF model [13]. Prior to conducting a clinical trial with ALF patients, we tried to identify a PD marker of rh-HGF, which could allow us to evaluate the pharmacological outcome of rh-HGF besides the conventional clinical parameters for ALF care, such as AST, ALT, and PT. Moreover, the availability of such a PD marker for healthy subjects may help to determine appropriate dosing regimens of rh-HGF in patients with ALF.

We firstly chose live animals rather than an in vitro cell system to discover PD markers because it might mimic the clinical setting in humans well, considering the bio-distribution of rh-HGF in the body. rh-HGF is known to preferentially distribute to the liver and with much less amount to the kidney and lung [17]. Therefore, we first harvested the murine livers to identify the candidates for PD markers after rh-HGF administration, and, at the same time we pre-filtered candidates by limiting these to the molecules which were present extracellularly or on the plasma membrane. In addition, we selected molecules which showed sustained expression from the early (8 h) to the late (24 h) timepoint to heighten their likelihood of being detected in the serum independent of their serum half-lives. As a result, murine *Apoa4* was found to be the highest (Table 1). Besides *Apoa4*, other genes such as *Pcsk9*, *Tnfrsf9*, and *Il6ra* that were identified using DNA microarray, were remarkably induced by rh-HGF, although the induction was only transient at 8 or 24 h in most cases, or the signal values, which may impact the protein levels, were relatively low. Since rh-HGF impacts cell growth and regeneration, the induction of cell-cycle- or growth-related genes such as *Ccnd1*, *ccne1*, and *Myc* was very obvious and they showed upregulation (data not shown). While these intracellular molecules may be less efficient as soluble PD markers, their upregulation confirmed the in vivo biological impacts of administered rh-HGF [2,18].

Thereafter, the induction of APOA4 was validated by using an in vitro human system utilizing primary cultured human hepatocytes (Figure 1). Consistent with the previous report that human APOA4 is expressed primarily in the small intestine [19], the baseline levels of *APOA4* mRNA were quite low in human hepatocytes. However, we found that rh-HGF induced *APOA4* mRNA in hepatocytes and stimulated the cells to produce APOA4 protein in a dose-dependent manner, mediated by c-Met (Figure 1). In this culture system, rh-HGF enhanced the APOA4 protein production by four- to five-fold compared to that of the control. Therefore, APOA4 is assumed to have high signal-to-noise ratio when human hepatocytes are treated with rh-HGF. If this observation is proven to be true in clinical settings, APOA4 will be a promising PD marker because of its assay sensitivity.

In order to confirm the adequacy of APOA4 as a PD marker of rh-HGF, the pharmacodynamic profile of APOA4 was evaluated in live animals; initially in normal mice, followed by mice with liver injury. In mice, APOA4 is known to be expressed in the small intestine and the liver [19,20,21]. This was also observed in our study, and the expression of murine *Apoa4* was detected in both the organs at baseline with relatively lower values in the liver (Figure 2). However, rh-HGF strongly induced *Apoa4* mRNA only in the liver at 8 h in a dose-dependent manner and the induction was sustained throughout the 24 h (Figure 2A) and such an induction of *Apoa4* mRNA was not detected in the small intestine (Figure 2B). This discrepancy might be due to preferential bio-distribution of rh-HGF to the liver and not to the small intestine [17]. Even though we cannot deny the possibility that intestinal *Apoa4* is a downstream molecule other than c-Met in the intestine, the observed finding at least proves that the liver is the primary organ impacted by administered rh-HGF in vivo. Thereafter, we focused on the induction of hepatic mRNA and serum APOA4 levels, particularly addressing the c-Met dependency. Notably, rh-HGF induced *Apoa4* mRNA (Figure 3A) in the liver was associated with a dose-dependent increase in serum APOA4 protein concentration (Figure 3B,C). Importantly, the induction of APOA4 was inhibited by a c-Met inhibitor PF-04217903 both at mRNA and protein levels (Figure 3A–C), therefore, it strongly suggested that APOA4 directly represents the pharmacological effects of rh-HGF as the downstream molecule of c-Met signaling.

Finally, the adequacy of APOA4 as a PD marker was assessed in murine ALF model, which was developed by Fas-mediated cell death using an anti-Fas antibody [5]. Consistent with the observation in normal mice, rh-HGF drastically induced serum APOA4 protein in a dose-dependent manner in mice with ALF (Figure 4). We previously reported that the anti-apoptotic effects of rh-HGF on hepatocytes were significantly associated with the suppression of intrahepatic hemorrhage in which preserved PT was the best parameter that strongly correlated with the decrease in intrahepatic hemorrhages [13]. Considering the PT kinetics after administering rh-HGF [13], the inductive increase in murine APOA4 protein in serum by rh-HGF (Figure 4) reciprocally correlated with the suppression of PT prolongation [13], which might further exhibit the feasibility of APOA4 as a PD marker of rh-HGF.

Here, we primarily identified APOA4, one of the downstream molecules of HGF-c-Met signaling, to be a soluble PD marker of rh-HGF. Interestingly, it has been demonstrated that APOA4 is not just a constituent of high-density lipoprotein but it exerts anti-oxidant and anti-inflammatory effects to protect liver damage in mice [10,11]. Notably, serum APOA4 seemed to increase in murine ALF over normal mice without exogenous rh-HGF (Figure 4A; rh-HGF, 0 mg/kg). Endogenously induced HGF may be responsible for this, although there may be other molecular events regulating APOA4 synthesis. Taken together, it is essential to investigate the clinical utility of serum APOA4 in future clinical trials with healthy subjects and ALF patients at least to evaluate the pharmacodynamic effect of the rh-HGF, E3112. In addition, such human data may help further understanding of how rh-HGF exerts its therapeutic action in clinical settings, not just by direct effect on liver cells but also via APOA4 protein.

## 4. Methods

### 4.1. Recombinant Human HGF

Recombinant human HGF (rh-HGF) was manufactured by Eisai Co., Ltd. (Tokyo, Japan), and its designated code is E3112 [13]. Manufacturing was proceeded according to the good manufacturing practices guidelines.

### 4.2. rh-HGF Administration to Normal Mice

Six-week-old male BALB/c mice (Charles River Japan, Kanagawa, Japan) received intravenous injections of rh-HGF at the doses of 0.5, 1.5, 5 or 15 mg/kg. Phosphate-buffered saline (PBS) was injected as the vehicle control (0 mg/kg). The mice were sacrificed to collect the left lateral lobe of the liver at 8 or 24 h after rh-HGF administration and collect blood at 24 h. Total RNA was isolated from the liver with RNeasy^®^ Mini Kit (Qiagen, Hilden, Germany) according to the manufacturer’s instructions. In a separate experiment, c-Met inhibitor PF-04217903 [22,23] was orally administered at a dose of 50 mg/kg 1 h before rh-HGF administration. All the experimental protocols were approved by the Institutional Animal Care and Use Committee and were performed according to the regulations on animal experimentation by Eisai Co., Ltd. (Tokyo, Japan). The mice were maintained in a specific pathogen-free (SPF) area.

### 4.3. DNA Microarray Analysis

The DNA microarray used in the study was SurePrint G3 mouse GE 8 × 60 K Microarray G4852A (Agilent Technologies, Santa Clara, CA, USA). Genes that showed a signal value of 100 or more in all the groups were primarily selected. Among them, genes that were assumed to exist as secreted or soluble proteins were extracted through the information from the annotations such as “Extracellular” or “Plasma membrane” in the Gene Ontology database.

### 4.4. Primary Culture of Human Hepatocytes

Non-cryopreserved human hepatocytes (Biopredic International, Rennes, France) were primarily seeded on a collagen I-coated 24-well plate at 3.8 × 10^5^ cells/well and incubated at 37 °C with 5% CO_2_ in an incubation medium of Williams E with GlutaMax (Basal Hepatic Cell Medium, Biopredic International) containing 4 μg/mL bovine insulin, 100 IU/mL penicillin, 100 μg/mL streptomycin, and 50 μmol/L hydrocortisone hemisuccinate (Additives for Hepatocyte Culture Medium, Biopredic International). Following the incubation period of 48 h, hepatocytes were treated with PF-04217903 (final concentration: 100 nmol/L) or incubation medium containing 0.05% dimethyl sulfoxide at an incubation volume (450 μL/well) on the 24-well plate at 37 °C with 5% CO_2_ for 1 h. Subsequently, rh-HGF (final concentrations: 100, 300, and 1000 ng/mL, 50 μL/well), or vehicle control (incubation medium, 50 μL/well) was added to each well (500 μL/well: incubation volume) and incubated at 37 °C with 5% CO_2_ for 24 h. Three batches of human non-cryopreserved hepatocytes were used in this study. Each batch of hepatocytes was independently cultured in each experiment. Hepatocytes were treated in triplicate for each experiment.

### 4.5. Reverse Transcription and Quantitative Real-Time Polymerase Chain Reaction (PCR)

After approximately 24 h of incubation with rh-HGF in the in vitro experiments, total RNA was isolated from the remaining hepatocytes with RNeasy^®^ Micro Kit (Qiagen) according to the manufacturer’s instructions. Quantitative PCR analyses of the cDNA samples were performed and 6-point relative standard curves were obtained in triplicate with TaqMan^®^ PCR primers and probes (TaqMan Gene Expression Assays, Applied Biosystems, Foster City, CA, USA) and TaqMan Fast Advanced Master Mix (Applied Biosystems) on a ViiA7 Real-time PCR System (Applied Biosystems). Relative standard curves for the target gene (human *APOA4*) and endogenous control (18S ribosomal RNA (rRNA)) were obtained by preparing serial dilutions of cDNA samples from the hepatocytes that were treated with 1000 ng/mL of rh-HGF. *APOA4* and 18S rRNA obtained from the cDNA samples were quantified by TaqMan probes (*APOA4*: Hs00166636_m1, 18S rRNA: Hs99999901_s1, Applied Biosystems). Cycle threshold (Ct) values were determined by ViiA7 software (version 1.2.4, Applied Biosystems). A 6-point relative standard curve of the median value from the triplicate analyses was used to determine the relative quantity of the target gene in each cDNA sample. Relative gene expression of *APOA4* mRNA and the expression of 18S rRNA was calculated with the median value from the triplicate analyses.

In the in vivo experiment, cDNA was synthesized with High-Capacity cDNA Reverse Transcription Kit with RNase Inhibitor (Applied Biosystems) by taking 1000 ng of total RNA. Quantitative PCR analyses of the cDNA samples and 6-point relative standard curves were conducted in triplicate by TaqMan^®^ PCR primers and probes (TaqMan Gene Expression Assays, Applied Biosystems) and TaqMan Fast Advanced Master Mix (Applied Biosystems) on a ViiA7 Real-time PCR System (Applied Biosystems). Murine *Apoa4* and hypoxanthine guanine phosphoribosyl transferase (*Hprt*) obtained from cDNA were quantified with TaqMan probes (*Apoa4*: Mm00431814_m1, *Hprt*: Mm03024075_m1, Applied Biosystems). Cycle threshold (Ct) values were determined by using ViiA7 software (version 1.2.4, Applied Biosystems). A 6-point relative standard curve from the median value of triplicate analyses was used to determine relative quantity of the target gene in each cDNA sample. Relative gene expression of *Apoa4* mRNA and the expression level of *Hprt* mRNA was calculated with median value of triplicate.

### 4.6. Murine ALF Model

Six-week-old male BALB/c mice (Charles River Japan) received intravenous injections of 0.23 mg/kg anti-Fas monoclonal antibody (Jo2; BD Pharmingen, San Diego, CA, USA) to develop the Fas-mediated ALF model. rh-HGF was injected intravenously at doses of 0.5, 1.5, 5, or 15 mg/kg one hour prior to the anti-Fas injection. PBS was injected as the vehicle control. All of the test materials were administered at a volume of 10 mL/kg. Five hours after anti-Fas antibody administration, the mice were sacrificed and their serum APOA4 levels in the collected blood samples were measured by Western blot.

### 4.7. Western Blot Analysis

APOA4 protein was analyzed by standard Western blot with SDS-PAGE. Murine APOA4 was detected by anti-murine APOA4 antibody (GeneTex, Irvine, CA, USA) followed by peroxidase-conjugated anti-goat secondary antibodies (Invitrogen, Carlsbad, CA, USA). Serum sample from the non-treated control calibration group was used to create a standard curve, and the levels of murine APOA4 were calculated along the standard curve. Human APOA4 was detected by anti-human APOA4 antibody (Proteintech Group, Chicago, IL, USA) followed by peroxidase-conjugated anti-rabbit secondary antibodies (Jackson ImmunoResearch Laboratories, West Grove, PA, USA). Purified recombinant human APOA4 proteins (KAN Research Institute, Hyogo, Japan) were diluted to 100, 50, 25, 12.5, and 6.25 ng/mL with PBS as standard. The concentrations of human APOA4 were calculated along the calibration curve. For both murine and human APOA4, secondary antibodies were visualized by SuperSignalTM West Dura Extended Duration Substrate (Thermo Fisher Scientific, Waltham, MA, USA).

### 4.8. Statistics

The data were presented as mean ± standard error of the mean (SEM). Statistical analyses were performed on GraphPad Prism v7.02 (GraphPad Software, La Jolla, CA, USA). *p* values < 0.05 were considered statistically significant.

## Figures and Tables

**Figure 1 ijms-22-04578-f001:**
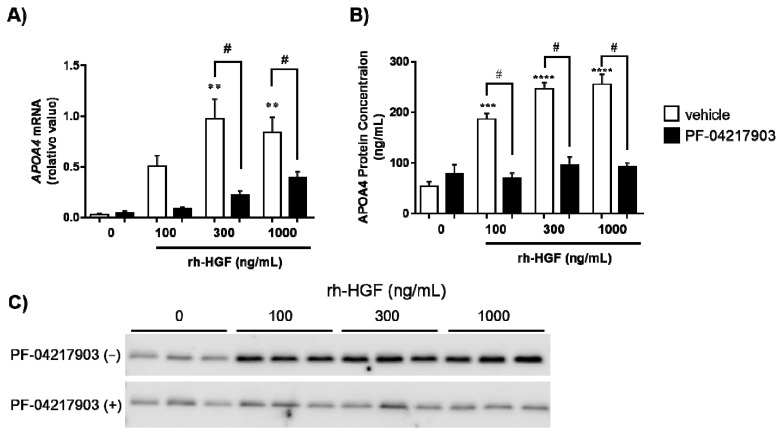
Induction of human APOA4 by rh-HGF, and its c-Met dependency. (**A**) Relative expression of *APOA4* mRNA compared to the expression of 18S rRNA in human hepatocytes, (**B**) Relative concentration of APOA4 protein in supernatants ** *p* < 0.01, *** *p* < 0.001, **** *p* < 0.0001: comparison between vehicle control (incubation medium) or rh-HGF alone-treated group (Dunnett’s test). ^#^
*p* < 0.05: comparison between rh-HGF alone-treated group and PF-04217903-rh-HGF co-culture group (unpaired *t* test). The data are presented as mean ± SEM (*n* = 3). The median value of the triplicate analyses was used for calculation. (**C**) Analysis of APOA4 protein levels in the supernatants of the culture using Western blot. The picture shows a representative batch of hepatocytes in triplicate.

**Figure 2 ijms-22-04578-f002:**
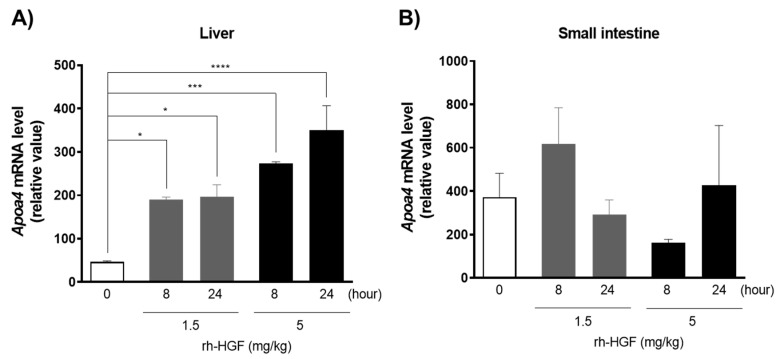
Expression of *Apoa4* mRNA in the liver and small intestine after rh-HGF administration in mice. (**A**) Liver, (**B**) Small intestine. Relative expression of *Apoa4* mRNA against the expression of *Hprt* mRNA was calculated using the median value of the triplicate analyses * *p* < 0.05, *** *p* < 0.001, **** *p* < 0.0001: comparison between the vehicle control (PBS)- and rh-HGF-treated groups (Dunnett’s test). The data are presented as mean ± SEM (*n* = 3).

**Figure 3 ijms-22-04578-f003:**
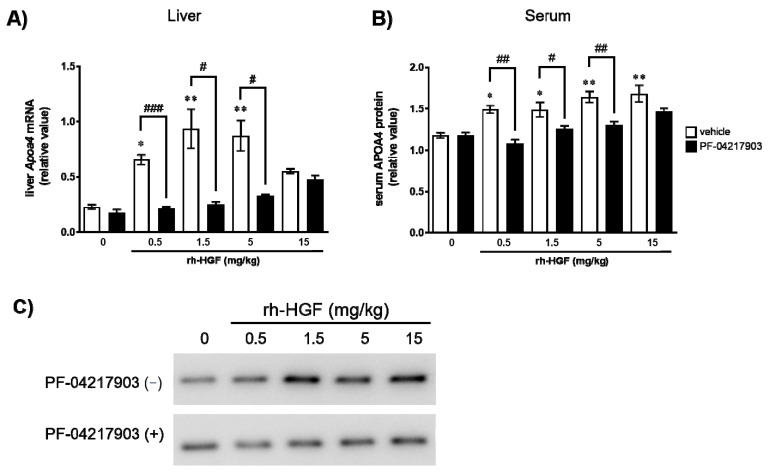
c-Met dependent induction of murine APOA4. Twenty-four hours after rh-HGF administration with or without PF-04217903 treatment, the mice were sacrificed, and their livers and bloods were collected for the analyses. (**A**) Relative expression of *Apoa4* mRNA in the liver compared with the expression of *Hprt* mRNA in mice, (**B**) Relative value of APOA4 protein in serum * *p* < 0.05, ** *p* < 0.01: comparison between vehicle control (PBS) and rh-HGF alone-treated groups (Dunnett’s test), ^#^
*p* < 0.05, ^##^
*p* < 0.01, ^###^
*p* < 0.001: comparison between rh-HGF alone-treated group and PF-04217903- rh-HGF co-administered group (unpaired *t* test). The data are presented as mean ± SEM (n = 3). The median value of the triplicate analyses was used for calculation. (**C**) Analysis of APOA4 protein levels in serum using Western blot. The picture shows a representative image of lanes showing individual animals.

**Figure 4 ijms-22-04578-f004:**
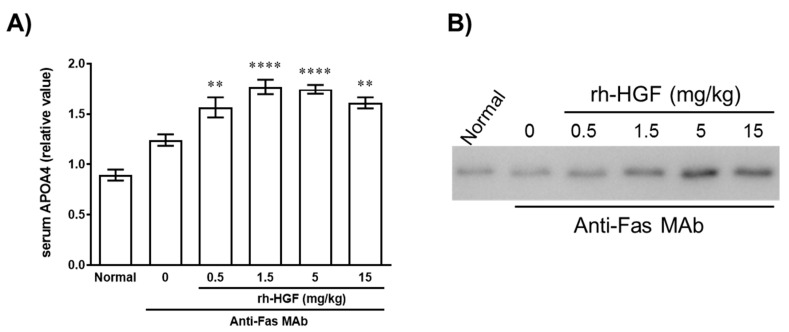
rh-HGF-mediated induction of APOA4 protein in anti-Fas-induced murine ALF model. rh-HGF was injected intravenously one hour prior to the anti-Fas injection. Five hours after anti-Fas administration, the mice were sacrificed and their blood collected for the analyses. (**A**) Relative value of APOA4 protein in serum. PBS was administered in the vehicle control group (rh-HGF, 0 mg/kg). ** *p* < 0.01, **** *p* < 0.0001: comparison between the vehicle control- and rh-HGF-treated groups (Dunnett’s test). The data are presented as mean ± SEM (*n* = 5–8). (**B**) A representative image of serum APOA4 levels by Western blot. Each lane shows individual animals.

**Table 1 ijms-22-04578-t001:** Genes highly induced by rh-HGF in the murine liver. Signal values at each timepoint show the average obtained from 3 mice which were sacrificed at designated timepoint before or after rh-HGF administration (1.5 mg/kg).

				Expression Change (versus 0 h)			
Gene Symbol	Signal Value (Average)			8 h		24 h	
	0 h	8 h	24 h	Ratio	*p* Value	Ratio	*p* Value
*Apoa4*	2436	15,738	13,453	6.46	0.0002	5.52	0.0016
*Prss8*	37	123	198	3.32	0.0955	5.35	0.0903
*Defa1*	34	37	106	1.09	0.8609	3.11	0.2846
*Tff3*	56	132	167	2.35	0.0623	2.97	0.1078
*Hmgb2*	701	553	1679	0.79	0.0879	2.40	0.0009
*Dnase1*	46	59	109	1.29	0.4958	2.39	0.2253
*Apoc3*	408	1059	938	2.60	0.0002	2.30	0.0004
*Afp*	71	349	148	4.89	0.0197	2.07	0.0170
*Spink3*	83	66	166	0.80	0.2707	2.01	0.2758
*Tnfrsf9*	38,813	112,574	74,826	2.90	0.0043	1.93	0.0192
*Il15ra*	44,941	118,655	83,474	2.64	0.0092	1.86	0.0151
*Pcsk9*	567	1458	1036	2.57	0.0187	1.83	0.0431
*Gast*	37	357	63	9.54	0.0005	1.69	0.1393
*Nrg4*	75	1724	112	22.85	0.0001	1.48	0.0421
*Il6ra*	2912	14,047	3742	4.82	0.0005	1.29	0.1408
*Fhad1*	124	1212	143	9.79	0.0003	1.16	0.0674
*Cgref1*	113	1350	128	11.96	0.0013	1.13	0.6870
*Alpl*	411	1499	456	3.64	0.0097	1.11	0.5636
*Fam3c*	1807	5240	1962	2.90	0.0002	1.09	0.2547
*Serpina7*	373	1636	283	4.38	0.0018	0.76	0.1721
*Creld2*	11,476	34,924	8179	3.04	0.0433	0.71	0.3436
*Cxcl14*	34	161	21	4.81	0.0073	0.62	0.3495

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
