# Peer review of "Serum APOA4 Pharmacodynamically Represents Administered Recombinant Human Hepatocyte Growth Factor (E3112)"

_ijms, 2021, doi:10.3390/ijms22094578_

Round 1

Reviewer 1 Report

Sotaro Motoi and colleagues studied the potential role of serum APOA4 as a pharmacodynamic marker of rh-HGF stimulation, after in vitro and in vivo experiments.

The results are original, and future studies are needed to evaluate the potential clinical role of this marker in humans in the setting of acute liver failure.

Major Comment:

No data, at least in my knowledge, exist about a possible relation among HGF and c-Met signaling and APOA4 induction, and the authors concluded that this serum marker could be used in clinical trial in humans.

Thus, author need to clarify mechanistically and with more details, as serum APOA4 protein is activated by HGF via C-MET signaling.

Minor Comments:

  • The authors in the last part of the introduction, from line 79 to line 87, instead of giving more information about the role of APOA4, resume the results of the study. This part is not consistent with the introduction and need to be revised.
  • Primers used for quantitative PCR analyses of the C-DNA samples (human and murine) should me mentioned.

Reviewer 2 Report

Some minor issues:

  1. Authors use random notations like Apoa4 and APOA4.
  2. Table 1 – how many microarrays were used for each group? Was it only one per group or more?
  3. 2B – authors observe huge differences in APOA4 mRNA expression in the intestine. A lower dose of rh-HGF seems to induce APOA4 expression (however it is not statistically significant), however, at a higher dose (5 mg/kg) a decrease has been observed. Still, not statistically significant, which is amazing taking into account the low SEM observed in this group. Maybe the number of observations that equals to 3 is too small to obtain some conclusive results?
  4. 4 is unclear. Fig. 4A represents relative APOA4 conc. in serum, however, authors didn’t show how it was calculated – relative to what protein? How many animals were used in each group? Does ‘normal’ mean control group?
  5. 4B – data based on one animal per group. I suppose it is only provided for general information, and the results are presented on fig. 4A?
  6. 408-409 – How standard curve has been created for murine APOA4 assessment? By dilution of control group serum authors could possibly create a standard curve with lower APOA4 concentrations, however, in all treated animals APOA4 concentrations were higher, so above the highest standard. Does it mean that the serum was additionally diluted before the assessment?
  7. For ALF model authors should provide some other markers of liver injury, to prove that anti-Fas Mab was effective in inducing hepatocyte apoptosis.

Round 2

Reviewer 1 Report

The paper is suitable for publication in the present form. 

Author Response

Thank you for your time